# Breast hypoplasia markers among women who report insufficient milk production: A retrospective online survey

Renee L. Kam[1]*, Lisa H. Amir[1,2], Meabh Cullinane[1], Jenny Ingram[3], Xia Li[4], Laurie A. Nommsen-Rivers[5]

1 Judith Lumley Centre, School of Nursing and Midwifery, La Trobe University, Bundoora, Victoria, Australia, 2 Breastfeeding Service, Royal Women's Hospital, Parkville, Victoria, Australia, 3 Bristol Medical School, University of Bristol, Bristol, United Kingdom, 4 Department of Mathematics and Statistics, La Trobe University, Bundoora, Victoria, Australia, 5 College of Allied Health Sciences, University of Cincinnati, Cincinnati, Ohio, United States of America

* renee.kam@latrobe.edu.au

## Abstract

### Objectives

To estimate the proportions of anatomical breast characteristics suggestive of breast hypoplasia among breastfeeding women self-reporting low milk supply. We also explored breast hypoplasia risk factors.

### Design

Online survey conducted between October 2021 and January 2022.

### Setting

**Five** low milk supply Facebook groups.

### Participants

487 women reporting low milk supply with their first child born $\geq$ 37 weeks gestation within 5 years of participation in this study, and residing in the USA, Australia or the UK. We present data on the primary outcome ('breast type') for 399 women. Women were excluded if the dyad was separated for more than 24 hours during the hospital stay, or if the mother reported removing milk less than 6 times per day from each breast on most days before being aware of having insufficient milk production.

### Primary and secondary outcome measures

The proportions of proposed breast hypoplasia markers including atypical breast type, widely spaced breasts, breast asymmetry, stretch marks on the breast and lack of pregnancy breast growth. We also estimated the odds of having breast hypoplasia markers in at-risk groups compared to reference groups, adjusting for covariates.

**Data Availability Statement:** Data cannot be shared publicly because we do not have approval from the La Trobe University Human Research Ethics Committee. The dataset is available from the

corresponding author upon reasonable request; or contact the La Trobe University Human Research Ethics Committee (humanethics@latrobe.edu.au).

**Funding:** The author(s) received no specific funding for this work.

**Competing interests:** The authors have declared no competing interests exist.

## Results

Approximately 68% reported at least one atypical breast (270/399; 95% CI: 62.9%, 72.1%). Around 47% reported widely spaced breasts (212/449; 95% CI: 42.7%, 52.7%), 72% a lack of pregnancy breast growth (322/449; 95% CI: 68.3%, 77.4%), and 76% stretch marks on the breast (191/250; 95% CI: 70.7%, 81.3%). Multiple logistic regression analyses identified being overweight during pubertal years as a risk factor for atypical breast type and lack of pregnancy breast growth.

## Conclusions

Participants in low milk supply Facebook groups reported high rates of breast hypoplasia markers. Being overweight during adolescence was a risk factor for breast hypoplasia markers. These findings should be confirmed in well-conducted large cohort studies to determine the strongest combination of hypoplasia markers in predicting low supply.

## Introduction

Although most new mothers commence breastfeeding, many cite low milk supply as the reason for stopping breastfeeding prematurely [1–3]. The proportion of women ceasing breastfeeding early who have a perceived versus an actual insufficient milk production is unknown. A perceived low supply occurs when a mother believes her supply is insufficient regardless of whether an actual low supply exists or not [4]. An actual low supply can result from breastfeeding challenges (secondary low supply) or can inherently exist (primary low supply) [5]. A mother experiences primary insufficient milk supply when her body cannot produce enough milk to enable exclusive breastfeeding despite regular milk removal [5]. One possible reason for primary insufficient milk production is breast hypoplasia [6]. Although the outward appearance of the breast is not conclusive, it is thought that breasts appearing as hypoplastic lack sufficient glandular tissue.

Breast hypoplasia can be congenital or acquired [7]. Congenital breast hypoplasia is associated with uncommon syndromes (e.g. Poland or Jeune), chest wall deformities (e.g. pectus excavatum), mitral valve prolapse, or hormonal disruption due to oestrogen insensitivity or endocrine disrupting chemicals [7–9]. In addition, there is increasing concern that exposure to environmental contaminants in utero or during puberty may impair mammary gland development [10]. Acquired breast hypoplasia can be associated with a history of breast radiation, breast reduction surgery or breast haemangioma [7]. Other acquired cases of breast hypoplasia have no identifiable cause, although pubertal and/or gestational glandular tissue development may be hampered by various endocrine alterations [11–20].

There is a lack of research investigating possible links between breast anatomical variations and lactation outcomes. Ventura et al found that among women with "more dense areolae", shorter and wider nipples were associated with a greater chance of experiencing low milk supply and slow infant weight gain [21]. A study by Vazirinejad and colleagues determined that infants of mothers with breast variations (any form of "large nipple", "flat nipple", "inverted nipple" and "abnormally large breast") had significantly lower weight gain than infants of mothers without these variations [22]. It is unclear if the poor breastfeeding outcomes were directly related to the anatomical issues or due to difficulties with infant latching to the breast and effectively removing milk. Other researchers found that no or slight pregnancy breast

growth, and no or slight postpartum breast engorgement with secretory activation (lactogenesis II or "milk coming in"), were associated with inadequate infant weight gain and shorter breastfeeding duration [11, 23].

Breast hypoplasia has been recognised as a characteristic of the 'tuberous' breast deformity in the plastic surgery literature since 1976 [24]. Prior to our research, the largest study to investigate a possible relationship between anatomical breast characteristics suggestive of breast hypoplasia and milk production was a case series of 34 women conducted by Huggins et al [25]. These researchers adapted a tool from a retrospective analysis of 40 patients undergoing operative breast corrections [26] to categorise women's breasts into one of four progressive types of tuberous breast (S1 Fig) [25]. In Huggins and colleagues' sample, the women's 'breast type' appeared to be related to the adequacy of their milk production, as women with type 2, 3 or 4 breasts produced insufficient milk compared to women with type 1 ("typical appearance") breasts [25].

Huggins and colleagues also identified other anatomical breast characteristics they suspected were associated with primary insufficient milk production due to breast hypoplasia [25]:

- Noticeable breast asymmetry (i.e., a marked difference in size, or size and shape, of the breasts);

- A wide intermammary width ($\geq$ 3.8 cm or 1.5 inches), because of underdevelopment of the inner aspect of the breast;

- Stretch marks on one or both breasts (the authors observed their presence when evaluating breast hypoplasia);

- Little or no pregnancy breast growth, which may suggest atypical mammogenesis;

- A lack of breast fullness in the first week postpartum which may indicate a deficiency in secretory differentiation in pregnancy and/or secretory activation (lactogenesis II) after giving birth.

No prior research has explored the prevalence of different breast types or proposed markers of breast hypoplasia among women with low milk production. Also, the feasibility of asking women to self-report their own anatomical breast characteristics has not previously been examined. Because of the importance of breastfeeding for maternal and infant health [27], it is important to elucidate the role maternal breast anatomy plays in low milk production. Therefore, using an online survey of women self-identifying as having low milk supply, we aimed to estimate the proportion of this sample of women with various anatomical breast characteristics related to breast hypoplasia, assess the feasibility of maternal self-report of these characteristics and to explore breast hypoplasia risk factors.

## Methods

### Objectives

**Primary objective.** The primary objective was to estimate the proportion of women with self-reported low milk production describing at least one breast as type 2, 3 or 4 as per Huggins et al (S1 Fig) [25]. In this paper we refer to a type 2, 3 or 4 breast as an 'atypical' breast type and a type 1 breast as a 'typical' breast type.

**Secondary objectives.** Secondary aims included determining the feasibility of asking women to self-report their breast anatomy characteristics and assessing the proportions of proposed markers of breast hypoplasia including a wide space between the breasts (referred to as

'intramammary space' by Huggins et al) ($\geq$ 1.5 inches or 3.8 cm) [25]; lack of breast growth during pregnancy with their first child (no noticeable change in, or an increase of < 1 bra cup size to either breast); presence of stretch marks on one or both breasts [25] prior to birth of first child; and breast asymmetry ($\geq$ 2 cup size difference between their breasts).

In addition, we aimed to determine the proportion of women with delayed secretory activation (breasts becoming noticeably fuller > 72 hours postpartum [28, 29] or breasts never became noticeably fuller with participants' first child).

We also planned to explore associations between endocrine conditions (polycystic ovary syndrome, diabetes, hypothyroidism) and BMI, and at least one breast being atypical; whether a link exists between the endocrine conditions listed above or BMI and proposed markers of breast hypoplasia (listed above); if proposed markers of breast hypoplasia are associated with breast type; and whether a relationship exists between having at least one atypical breast and delayed secretory activation.

## Design

We conducted an open voluntary retrospective online survey of women belonging to low milk production Facebook support groups [30, 31]. This design enabled us to recruit participants and conduct research at a time when face-to-face research was limited due to the COVID-19 pandemic. It also enabled timely recruitment of participants in our target population: women with low milk supply. This study received approval by the La Trobe University Human Ethics Committee (approved 21 September 2021; approval number HEC21306).

## Sample and eligibility criteria

A convenience sample of women who self-reported low milk supply completed the survey. No incentives were offered for participation. Women were eligible for participation if they typically resided in Australia, the United States or the United Kingdom; were 18 years or older; could read and write in English; and reported low milk supply with their first live birth of a term singleton ($\geq$ 37.0 weeks gestation) born within 5 years of participation in the study. In order to reduce secondary causes of low milk production, exclusion criteria included separation of the mother/infant dyad for more than 24 hours during their hospital stay after the birth; or if the mother reported not removing milk at least 6 times per day from each breast on most days prior to being aware of having insufficient milk production with the first infant.

## Patient and public involvement

While community members were not directly involved in designing or conducting this project, the investigators used their experience in caring for women with low milk supply (RLK, LHA, JI) and working with research participants with low milk supply (LNR) to inform the research questions and analyses.

## Survey

Previously, we devised a survey and diagram depicting breast types to conduct a reliability study and confirmed that researchers could reliably measure women's intermammary width [32]. The data collection tool for this current study was designed by adapting items from the survey used in the reliability study [32], with the addition of several new questions investigating anatomical breast characteristics. The breast type classification (primary outcome) item was based on the breast type diagram devised for the reliability study [32]. All survey questions

were piloted with the research team (n = 5), research colleagues (n = 10) and a small group of women who had low milk supply (n = 7) in an iterative manner.

The survey was launched and shared in low milk supply Facebook groups as well as via the first author's personal and business (lactation consultant) Facebook pages. Keywords used to search Facebook for low milk supply support groups included "breast hypoplasia", "insufficient glandular tissue", "supply line" or "low milk supply". The five low milk supply Facebook support groups where the survey was shared were: 'IGT And Low Milk Supply Support Group', 'Supply Line Breastfeeders Support Group of Australia', 'IGT Off Topic Group', 'Low Milk Supply—A Mother's Love' and 'Low Milk Supply/Domperidone' (S1 Table). The first author joined each group and contacted the group administrator(s) to provide information about the study and request permission to recruit participants using the Facebook site.

The survey consisted of 78 items organised into a structured online questionnaire with skip logic, and was administered through REDCap, a secure web-based application for data collection and management (S1 File) [33, 34]. All questions related to participants' first child. Where relevant, questions had "unsure" and "prefer not to say" as options. Depending on skip logic, the survey was up to 21 pages with up to 11 questions per page. Participants could go back to a previous page to review or change responses if they wished.

Women were screened using an eligibility survey (S2 File) where they were informed about the purpose of the study. Eligible participants could download and read the Participant Information Statement and were asked "Do you agree to complete the survey? Clicking 'Yes' tells us you want to take part in the study." If a participant provided consent to complete the survey by clicking 'Yes', they were led to the survey. We had access to no information that could identify individual participants during or after data collection.

The survey was open for 16 weeks between October 2021 and January 2022. When the invitation to participate in the study was first posed on each Facebook group, the first author or group administrator provided a brief description of the study purpose and a link to the REDCap survey. Snowballing was possible as the post may have been shared with other Facebook groups/pages/members. The first author interacted by thanking group members for their participation to help maintain traffic to the posts. Additional posts, again describing study purpose and linking to survey, were made 1–2 more times in the largest two groups ('IGT and Low Milk Supply Support Group' and 'Supply Line Breastfeeders Support Group of Australia') over the recruiting period until the target sample size was reached.

### Variables

**Outcome variables.**   The primary outcome was the proportion of participants with at least one atypical breast. Participants were asked to indicate what each of their breasts individually looked like just prior to pregnancy with their first child using S2 Fig. Participants who had breast surgery prior to the birth of their first child were asked to report breast appearance prior to surgery.

A secondary outcome was the feasibility of self-administration of the survey based on the proportion for whom breast type and other individual markers suggestive of breast hypoplasia could be determined. Additional secondary outcomes included the proportion of participants with other individual markers suggestive of breast hypoplasia as well as a delay in secretory activation. We also estimated the odds of having markers of breast hypoplasia in at-risk groups compared to reference groups (e.g., normal BMI, absence of disorder), adjusting for covariates.

**Exposure variables.**   Various endocrine alterations including polycystic ovary syndrome (PCOS), diabetes (type I, II or gestational) and hypothyroidism have been identified as being

associated with breast hypoplasia [11–20]. Therefore, participants were asked whether they had any such endocrine conditions medically diagnosed prior to the birth of their first child. Data were collected about the timing of onset of these conditions and medications used to manage a diagnosis of PCOS, gestational diabetes mellitus (GDM) or type II diabetes.

Participants were asked to describe their weight between 8 and 20 years of age using the following categories: 'underweight', 'normal weight', 'a little overweight', 'moderately overweight', 'very overweight', 'unsure', 'prefer not to say' or 'other'. We refer to this variable as 'youth weight'.

We asked the participants to provide estimates of their height and weight just before pregnancy with their first child in order to calculate their pre-pregnancy BMI. BMI was defined as per the World Health Organization BMI categories [35]: $<18.5$ kg/m$^2$ (underweight), 18.5 to $<25.0$ kg/m$^2$ (normal weight), 25.0 to $<30.0$ kg/m$^2$ (overweight), 30.0 to $<35.0$ kg/m$^2$ (obese class 1), 35.0 to $<40.0$ kg/m$^2$ (obese class 2), and $\geq40.0$ kg/m$^2$ (obese class 3).

**Other covariates.** Demographic characteristics including current age, country of residence, marital status and education were collected. Ethnicity was collected separately for each country where eligible women usually resided in. For participants who typically resided in Australia, questions related to indigeneity and country of birth were asked. Data about intention to breastfeed (by asking how long women planned to breastfeed their baby for) were collected.

Participants were also asked about medical conditions and obstetric history as these covariates may influence lactation outcomes. A final open-ended question was asked about participants' personal stories of how their low milk supply was discovered or diagnosed (not included in this paper).

## Sample size

Sample size was calculated to estimate the proportion of participants with at least one atypical breast [36]. *A priori*, we estimated the proportion of women with at least one atypical breast to be 50%. To ensure the 95% confidence interval (CI) estimate of the proportion of women who report low milk supply with at least one atypical breast is within 5% of the true population proportion, a sample of 385 was needed. Accounting for a 20% incomplete survey response, we aimed to recruit 482 women.

## Statistical analyses

**Primary outcome.** The estimated proportion of women in our sample having at least one atypical breast and the 95% CI around the estimate was determined. The numerator was based on the total number of participants coded 'atypical' and the denominator represented the sum of participants coded as 'typical' plus 'atypical' based on their responses. "None", "unsure" and missing responses were excluded from the primary result; in sensitivity analysis, we included these responses in the denominator to determine the potential impact of their missingness on the estimated prevalence of at least one atypical breast in this population.

**Secondary outcomes.** The feasibility of collecting information directly from women using an online survey was measured by calculating the proportion of respondents definitively answering the items related to the primary and secondary outcomes, compared to the proportion who skipped answering these items or indicated 'unsure.' Participants' open text responses were examined to identify any indication of confusion or feedback about these items.

The proportion of participants with proposed markers of breast hypoplasia was estimated and 95% CI calculated. "Unsure" and missing responses were not included in these analyses.

The chi-square ($\chi$2) test was used to examine bivariate associations between exposure and outcome variables. Effect sizes were determined using Cramer's V. For associations where p<0.10, multiple logistic regression was used to estimate the odds of the outcome in the at-risk group compared to the reference group, adjusting for covariates in a progressive manner.

We performed all statistical analyses in Stata version 15 [37]. The significance level used was p<0.05. Reporting for this study followed the Checklist for Reporting Results of Internet E-Surveys (CHERRIES) statement (S3 File) [30].

## Results

A total of 487 participants commenced the survey; 399 responded to our primary outcome (breast type) (81.9%). Of participants who responded to the breast type question, 67.9% resided in the United States of America, 23.3% in Australia and 8.8% in the United Kingdom (Table 1). The mean age of participants was 32 years (SD 4.6) and most (84%) had either a bachelor or postgraduate degree. The majority (85.1%) of participants intended to breastfeed for at least 12 months (320/376). Socio-demographic characteristics based on data from participants who responded to the breast type question are summarised in Table 1.

Chi-square analyses between sociodemographic characteristics displayed in Table 1 and breast characteristics which demonstrated associations where p<0.10 included breast type and i) age ($\chi^2$(1) = 6.2999, p = 0.012) and ii) country of residence ($\chi^2$(1) = 8.3689, p = 0.004); widely spaced breasts and i) country of residence ($\chi^2$(1) = 7.6443, p = 0.006) and ii) USA ethnicity ($\chi^2$(1) = 2.7848, p = 0.095); asymmetry and UK ethnicity ($\chi^2$(1) = 9.4138, p = 0.002); lack of growth in pregnancy with first child and i) age ($\chi^2$(1) = 3.15078, p = 0.076) and ii) country of residence ($\chi^2$(1) = 2.7586, p = 0.097); presence of stretch marks prior to birth of first child and USA ethnicity ($\chi^2$(1) = 4.4058, p = 0.036).

### Primary outcome

Around two thirds (67.7%) of participants reported <u>at least one</u> atypical breast (270/399; 95% CI: 62.9, 72.1). Most women (328/394; 83%) reported both their breasts were the same type (Table 2).

The steps taken to determine the denominator for the primary outcome calculation were:

i.  394/487 participants recorded responses to both questions about their right and left breast types.

ii.  Another two participants responded about their right but not their left breast type. These two participants were therefore included in the denominator for the primary outcome (i.e., 394+2 = 396).

iii.  Three additional responses were included in the denominator due to re-coding (i.e., when one breast had been identified as either typical or atypical and the other as missing, "unsure" or "none"; i.e., 396+3 = 399).

The denominator for the primary outcome calculation does not include 88 responses which were missing (n = 77), "unsure" (n = 5) or "none" (n = 4) for both breasts. Most of these missing responses (51/77; 66%) were due to branching logic error in the REDCap survey which was rectified once identified.

### Secondary outcomes

Survey participants were able to comprehend the survey items about markers suggestive of breast hypoplasia, with over 80% responding to these items (S2 Table). Only one open text

**Table 1. Sociodemographic characteristics of participants\*.**

| Characteristic | United States of America (N = 271) | Australia (N = 93) | United Kingdom (N = 35) | Overall (N = 399) |
|---|---|---|---|---|
| Age (years) | | | | |
| Mean (SD) | 33 (4.6) | 34 (4.3) | 35 (3.8) | 32 (4.6) |
| Median [Min; Max] | 33 [22;48] | 34 [25; 44] | 34 [28; 42] | 33 [22;48] |
| Missing | 12 | | 4 | 16 |
| Marital status, n (%) | | | | |
| Married or living with partner | 262 (96.7) | 91 (97.8) | 35 (100) | 388 (97.2) |
| Single | 6 (2.2) | 2 (2.2) | | 8 (2.0) |
| In a relationship but not living together | 3 (1.1) | | | 3 (0.8) |
| Highest level of educational attainment n (%) | | | | |
| Postgraduate degree | 108 (39.9) | 38 (40.9) | 14 (40.0) | 160 (40.1) |
| Bachelor degree | 113 (41.7) | 43 (46.2) | 19 (54.3) | 175 (43.9) |
| Some training beyond secondary school / secondary school | 50 (18.5) | 12 (12.9) | 2 (5.7) | 64 (16.0) |
| **Australian participants** | | | | |
| Birth country n (%) | | | | |
| Australia | | 78 (83.9) | | |
| United Kingdom | | 6 (6.5) | | |
| New Zealand | | 3 (3.2) | | |
| Other\* | | 6 (6.5) | | |
| Aboriginal or Torres Strait Islander n (%) | | | | |
| No | | 90 (96.8) | | |
| Yes | | 3 (3.2) | | |
| **USA participants** | | | | |
| Ethnic group[†] n (%) | | | | |
| White | 241 (88.9) | | | |
| Hispanic/Latina | 21 (7.7) | | | |
| Asian | 16 (5.9) | | | |
| Black, African or Caribbean | 4 (1.5) | | | |
| Other\* | 1 (0.4) | | | |
| **UK participants** | | | | |
| Ethnic group[†] n (%) | | | | |
| White | | | 32 (91.4) | |
| Black, African, Caribbean or Black British | | | 2 (5.7) | |
| Asian or Asian British | | | 1 (2.9) | |
| Other[‡] | | | 1 (2.9) | |

\*Of participants who responded to breast type (primary outcome) survey question

[†]Participants could choose one or more categories so total percentage does not equal 100

[‡]Other: Australian participants–USA (1), Chile (1), France (1), Tajikistan (1); USA participants–Middle Eastern (1); UK participants–Ashkenazi Jewish (1)

response mentioned a lack of clarity about the wording of cup size difference item. We hypothesised that women with higher BMI might have more difficulty categorising their breasts compared to women with lower BMI, but this was not confirmed. No association was found between missing data status by BMI category ($<25.0$ v $\geq 25.0$ kg/m$^2$, $\chi^2(1) = 0.0006$, $p = 0.981$)). A BMI of 25 was chosen as the reference level here because the World Health Organization categorises a BMI between 18.5 and $<25$ as normal weight [35].

**Table 2. Number of survey participants who indicated their breast anatomy according to Huggins' "breast type" [25]\*.**

|  | Right type 1 | Right type 2 | Right type 3 | Right type 4 | Total |
|---|---|---|---|---|---|
| Left type 1 | 113 | 12 | 2 | 1 | 128 |
| Left type 2 | 13 | 179 | 6 | 6 | 204 |
| Left type 3 | 0 | 8 | 18 | 3 | 29 |
| Left type 4 | 1 | 10 | 4 | 18 | 33 |
| Total | 127 | 209 | 30 | 28 | 394 |

\*Missing, "none" and "unsure" responses not included in this table

Darker shading: Participants reporting same breast type for each breast

Approximately 47% (212/449; 95% CI: 42.7%, 52.7%) of participants reported widely spaced breasts, and 72% noticed a lack of breast growth during pregnancy with their first child (322/449; 95% CI: 68.3%, 77.4%) (Table 3).

In our sample, 86.6% of participants (353/408; 95% CI: 82.8%, 89.5%) reported a delay or absence in secretory activation.

Based on participants who provided a response to the breast type (primary outcome) item (and excluding missing data), before the birth of their first child, 17.9% of our sample (69/386) reported having PCOS, 12.2% (48/392) reported hypothyroidism, 12.0% (47/392) reported GDM, 0.5% (2/395) reported type I diabetes, and 0.3% (1/394) reported type II diabetes.

Of participants who provided responses to the breast type (primary outcome) item (excluding missing data), most had a BMI in the overweight or obese category (24.1% (94/390) reported being overweight, 20.7% (79/390) were obese class 1, 8.7% (34/390) were obese class 2, and 7.7% (30/390) reported being in obese class 3). Approximately 1% (5/390) were underweight and 38.0% (148/390) reported a normal weight. In our sample, 40.3% (108/268), 28.1% (25/89) and 30.3% (10/33) of US, Australian and UK participants respectively were classified as obese.

Using chi-square analyses, we explored bivariate relationships between suggested markers of breast hypoplasia, BMI, youth weight, PCOS, GDM and hypothyroidism and the presence of at least one atypical breast (Table 4). Women with a high BMI ($\geq 25$ kg/m$^2$) were more likely to report atypical compared to typical breasts (Cramer's V = 0.1487 [small effect size] [38], p = 0.036). Women who described being overweight between 8 and 20 years of age were more

**Table 3. Proportion of participants with markers suggestive of breast hypoplasia.**

| Proposed breast hypoplasia marker\* | % (95% CI) |
|---|---|
| Widely spaced (n = 212/449) | 47.2 (42.7, 52.7) |
| Asymmetry (n = 34/448) | 7.6 (5.0, 10.3) |
| Lack of growth (n = 322/449) | 71.7 (68.3, 77.4) |
| Stretch marks prior to birth of first child [t] (n = 191/250) | 76.4 (70.7, 81.3) |
| Stretch marks appeared between 8 and 20 years of age[‡] (n = 153/168) | 91.1 (85.7, 94.6) |
| Stretch marks appeared during pregnancy with their first child [‡] (n = 14/186) | 7.5 (4.5, 12.3) |

\*Based on all responses to these items, excluding missing and unsure responses. Widely spaced, intermammary width > 1.5 inches or 3.8 cm; Asymmetry, $\geq 2$ cup size difference between breasts; Lack of growth, lack of breast growth during pregnancy defined as no noticeable change in or an increase of < 1 bra cup size to either breast during pregnancy with their first child

[t]Of those who responded 'yes' to presence of stretch marks

[‡]Of those who responded 'yes' to presence of stretch marks and 'yes' to them appearing before the birth of first child

**Table 4. Relationship between breast type and other characteristics.**

| Characteristic* | n (%) with at least one atypical breast[†] | | n (%) without at least one atypical breast | | $\chi^2$, p value[‡] |
|---|---|---|---|---|---|
| **Metabolic health characteristic** | | | | | |
| PCOS | | | | | 3.7841, 0.052 |
| Yes (n = 69) | 54 | (78.3) | 15 | (21.7) | |
| No (n = 317) | 210 | (66.3) | 107 | (33.8) | |
| Hypothyroidism | | | | | 2.0268, 0.155 |
| Yes (n = 48) | 37 | (77.1) | 11 | (22.9) | |
| No (n = 344) | 230 | (66.9) | 114 | (33.1) | |
| GDM during pregnancy with first child | | | | | 1.6719, 0.196 |
| Yes (n = 47) | 36 | (76.7) | 11 | (23.4) | |
| No (n = 345) | 232 | (67.3) | 113 | (32.8) | |
| Pre-pregnancy BMI, kg/m$^2$ (n = 385)˜ | | | | | 8.5176, **0.036** |
| Normal weight (n = 148) | 88 | (59.5) | 60 | (40.5) | |
| Overweight (n = 94) | 68 | (72.3) | 26 | (27.7) | |
| Obese class 1 (n = 79) | 60 | (75.9) | 19 | (24.1) | |
| Obese class 2+(n = 64) | 46 | (71.9) | 18 | (28.1) | |
| Youth weight (n = 391˚ | | | | | 30.0112, <**0.001** |
| Normal (n = 140) | 75 | (53.6) | 65 | (46.4) | |
| Little overweight (n = 115) | 87 | (75.7) | 28 | (24.4) | |
| Moderately or very overweight (n = 108) | 91 | (84.3) | 17 | (15.7) | |
| **Proposed breast hypoplasia marker˜** | | | | | |
| Widely spaced | | | | | 79.4987, <**0.001** |
| Yes (n = 185) | 166 | (89.7) | 19 | (10.3) | |
| No (n = 203) | 96 | (47.3) | 107 | (52.7) | |
| Asymmetry | | | | | 0.1452, 0.703 |
| Yes (n = 28) | 18 | (64.3) | 10 | (35.7) | |
| No (n = 357) | 242 | (67.8) | 115 | (32.2) | |
| Stretch marks | | | | | 2.2858, 0.131 |
| Yes (n = 162) | 115 | (71.0) | 47 | (29.0) | |
| No (n = 55) | 33 | (60.0) | 22 | (40.0) | |
| Lack of growth | | | | | 25.7740, <**0.001** |
| Yes (n = 281) | 213 | (75.8) | 68 | (24.2) | |
| No (n = 108) | 53 | (49.1) | 55 | (50.9) | |

*This column includes numbers of participants for which we have data on breast type in addition to the characteristic indicated

[†]An 'atypical' breast is a type 2, 3 or 4 breast

[‡]Pearson chi-square

˜Widely spaced, intermammary width > 1.5 inches or 3.8 cm; Asymmetry, ≥ 2 cup size difference between breasts; Stretch marks, stretch marks on one or both breast/s prior to first child; Lack of growth, lack of breast growth during pregnancy defined as no noticeable change in or an increase of < 1 bra cup size to either breast during pregnancy with their first child

˜Normal weight, 18.5 to <25.0 kg/m$^2$; overweight, 25.0 to <30.0 kg/m$^2$; obese class 1, 30.0 to <35.0 kg/m$^2$; obese class 2+, ≥35.0 kg/m$^2$. Underweight category excluded due to inadequate sample size (n = 5). Obese categories 2 and above combined due to small sample sizes (n = 39 for obese 2 and n = 35 for obese 3)

˚Youth weight, description of weight between 8 and 20 years of age. Underweight excluded due to inadequate sample size (n = 16). Moderately and very overweight categories combined (n = 76 for moderately overweight and n = 32 for very overweight)

BMI, body mass index; GDM, gestational diabetes mellitus; PCOS, polycystic ovary syndrome

likely to report atypical compared to typical breasts (Cramer's V = 0.2875 [medium effect size], p<0.001). Women with widely spaced breasts or lack of pregnancy breast growth were also significantly more likely to report atypical breasts (Cramer's V = 0.4527 [medium effect size] and -0.2574 [medium effect size] respectively, p<0.001). Our sample provides some evidence that women with PCOS (n = 69) are more likely to report atypical breasts (Cramer's V = 0.0990 [small effect size], p = 0.052).

Using chi-square analyses, we explored relationships between endocrine conditions (including BMI and youth weight) and suggested markers of breast hypoplasia (Table 5). Women with a high BMI were more likely to have widely spaced breasts, stretch marks present on their breasts and lack of pregnancy breast growth (Cramer's V = 0.1870 [small effect size], p = 0.002; Cramer's V = 0.2150 [small effect size], p = 0.01; Cramer's V = 0.2250 [small effect size], p<0.001 respectively). Women who described being overweight between 8 and 20 years of age were more likely to have widely spaced breasts and a lack of pregnancy breast growth (Cramer's V = 0.1867 [small effect size], p = 0.001; Cramer's V = 2123 [small effect size], p<0.001) respectively). Also, women with PCOS were more likely to have stretch marks present on their breasts (Cramer's V = 0.1429 [small effect size], p = 0.026).

Women who reported a delay or absence in secretory activation were more likely to report at least one atypical breast (71%; 216/304) compared to women without at least one atypical breast (29%; 88/304) ($\chi^2(1)$ 4.1122, p = 0.043) (S3 Table).

We performed regression analyses on the four outcomes for which chi-square analyses had at least one predictor variable with p-value <0.10 (see Tables 4 and 5). Crude and adjusted odds ratios were obtained for these relationships by performing logistic regression analyses in stages. In the first stage of adjusted analyses, we adjusted for significant socio-demographic variables and current metabolic health variables. In the second stage we added in youth weight status to determine its direct effect on each outcome independent of current metabolic health variables.

Although BMI category was a significant predictor of atypical breasts in the unadjusted analysis, it was no longer significant after adjusting for age, country of residence, and PCOS status (Table 6, Model 1). However, youth weight category remained a strong predictor of atypical breasts in a model adjusted for all of these covariates (Table 6, Model 2). The odds of having at least one atypical breast were 2.96 (95% CI: 1.58, 5.56) and 6.14 (95% CI: 2.74, 13.72) times higher among women who described being a 'little overweight' and 'moderately or very overweight' between 8 and 20 years of age, respectively, compared to women without at least one atypical breast (adjusting for age, country of residence, PCOS status and BMI) (Table 6, Model 2).

Logistic regression modelling between various metabolic health exposures and widely spaced breasts, stretch marks on the breast and lack of pregnancy breast growth were also undertaken (S4–S6 Tables). BMI category was a significant predictor of widely spaced breasts, stretch marks on the breast and lack of pregnancy growth in the unadjusted analyses (S4 Table, Crude OR). However, it only remained a significant predictor for widely spaced breasts in the obese 1 category after adjusting for country of residence and USA ethnicity (S4 Table, Model 1). Also, BMI was no longer a significant predictor for stretch marks on the breast after adjusting for USA ethnicity and PCOS status (S5 Table, Model 1).

Although BMI category was a significant predictor of lack of pregnancy breast growth in the unadjusted analysis, it was no longer significant after adjusting for age, country of residence, and GDM status (S6 Table, Model 1). However, the moderately / very overweight youth weight category remained a strong predictor of lack of pregnancy breast growth in a model adjusted for all of these covariates (S6 Table, Model 2). Among women who described being 'moderately or very overweight' between 8 and 20 years of age, the odds of lack of pregnancy

**Table 5. Relationship between metabolic health characteristics and markers suggestive of breast hypoplasia.**

| Characteristics present prior to or during pregnancy with their first child* | Widely spaced[t] | | Asymmetry[t] | | Stretch marks[t] | | Lack of growth[t] | |
|---|---|---|---|---|---|---|---|---|
| | Yes n (%) | No n (%) | Yes n (%) | No n (%) | Yes n (%) | No n (%) | Yes n (%) | No n (%) |
| PCOS | | | | | | | | |
| Yes | 44 (55.7) | 35 (44.3) | 7 (9.0) | 71 (91.0) | 50 (87.7) | 7 (12.3) | 54 (69.2) | 24 (41.4) |
| No | 162 (45.8) | 192 (54.2) | 27 (7.7) | 324 (92.3) | 136 (73.5) | 49 (26.5) | 261 (72.5) | 99 (27.5) |
| p value[‡] | | 0.110 | | 0.705 | | **0.026** | | 0.560 |
| Hypothyroidism | | | | | | | | |
| Yes | 28 (51.9) | 26 (48.1) | 2 (3.9) | 49 (96.1) | 25 (80.6) | 6 (19.4) | 41 (75.9) | 13 (24.1) |
| No | 182 (47.3) | 203 (52.7) | 31 (8.1) | 354 (91.9) | 165 (73.4) | 51 (23.6) | 278 (71.1) | 113 (28.9) |
| p value[‡] | | 0.528 | | 0.405[§] | | 0.599 | | 0.461 |
| GDM | | | | | | | | |
| Yes | 29 (53.7) | 25 (46.3) | 5 (9.3) | 49 (90.7) | 25 (80.6) | 6 (19.4) | 43 (82.7) | 9 (17.3) |
| No | 181 (47.0) | 204 (53.0) | 29 (7.6) | 353 (92.4) | 163 (76.2) | 51 (23.8) | 276 (70.2) | 117 (29.8) |
| p value[‡] | | 0.357 | | 0.594[§] | | 0.581 | | 0.061 |
| Type I diabetes | | | | | | | | |
| Yes | 2 (66.7) | 1 (33.3) | 0 (0) | 3 (100) | 1 (100) | 0 (0) | 2 (66.7) | 1 (33.3) |
| No | 209 (47.6) | 230 (52.4) | 34 (7.8) | 402 (92.2) | 189 (76.8) | 57 (23.2) | 319 (71.7) | 126 (28.3) |
| p value[¯] | | 0.608 | | 1.000 | | 1.000 | | 1.000 |
| Type II diabetes | | | | | | | | |
| Yes | 1 (50.0) | 1 (50.0) | 0 (0) | 1 (100) | 0 (0) | 0 (0) | 1 (100) | 0 (0) |
| No | 211 (48.0) | 229 (52.0) | 34 (7.8) | 403 (92.2) | 190 (76.9) | 57 (23.1) | 319 (71.5) | 127 (28.5) |
| p value[¯] | | 1.000 | | 1.000 | | insuff obs | | 1.000 |
| Pre-pregnancy BMI[¯] | | | | | | | | |
| Normal weight | 57 (35.6) | 103 (64.4) | 10 (6.3) | 150 (93.8) | 38 (63.3) | 22 (36.7) | 97 (59.1) | 67 (40.9) |
| Overweight | 60 (54.1) | 51 (45.9) | 11 (9.6) | 103 (90.4) | 48 (73.8) | 17 (26.2) | 88 (77.9) | 25 (22.1) |
| Obese class 1 | 51 (58.0) | 37 (42.0) | 6 (7.2) | 77 (92.8) | 55 (87.3) | 8 (12.7) | 66 (75.9) | 21 (24.1) |
| Obese class 2+ | 37 (50.7) | 36 (49.3) | 7 (9.5) | 67 (90.5) | 47 (82.5) | 10 (17.5) | 63 (85.1) | 11 (14.9) |
| p value[‡] | | **0.002** | | 0.711 | | **0.01** | | **<0.001** |
| Youth weight[°] | | | | | | | | |
| Normal | 63 (39.4) | 97 (60.6) | 12 (7.6) | 146 (92.4) | 43 (65.2) | 23 (34.9) | 98 (61.6) | 61 (38.4) |
| Little overweight | 62 (50.0) | 62 (50.0) | 13 (10.4) | 112 (89.6) | 62 (75.6) | 20 (24.4) | 99 (76.7) | 30 (22.3) |
| Moderately or very overweight | 75 (62.0) | 46 (38.0) | 8 (6.6) | 114 (93.4) | 76 (84.4) | 14 (15.6) | 103 (83.7) | 20 (16.3) |
| p value[‡] | | **0.001** | | 0.516 | | **0.02** | | **<0.001** |

*This column includes numbers of participants for which we have data on each individual marker suggestive of breast hypoplasia in addition to the characteristic indicated

[t]widely spaced, intermammary width > 1.5 inches or 3.8 cm; asymmetry, ≥ 2 cup size difference between breasts; stretch marks, stretch marks on one or both breast/s prior to first child; lack of growth, lack of breast growth during pregnancy defined as no noticeable change in or an increase of < 1 bra cup size to either breast during pregnancy with their first child

[‡]Pearson chi-square

[¯]Fisher's exact

[¯]Normal weight, 18.5 to <25.0 kg/m$^2$; overweight, 25.0 to <30.0 kg/m$^2$; obese class 1, 30.0 to <35.0 kg/m$^2$; obese class 2+, ≥35.0 kg/m$^2$. Underweight category excluded due to inadequate sample size (n = 5). Obese categories 2 and above combined due to small sample sizes (n = 39 for obese 2 and n = 35 for obese 3)

[°]Youth weight, description of weight between 8 and 20 years of age. Underweight excluded due to inadequate sample size (n = 16). Moderately and very overweight categories combined (n = 76 for moderately overweight and n = 32 for very overweight)

BMI, body mass index kg/m$^2$; GDM, gestational diabetes mellitus; insuff obs, insufficient observations; PCOS, polycystic ovary syndrome

**Table 6. Logistic regression modelling of risk factors for atypical breast shape.**

| Metabolic characteristic | Reference category | Crude OR (95% CI) | Model 1 AOR* (95% CI) | Model 2 AOR[t] (95% CI) |
|---|---|---|---|---|
| **PCOS** | No PCOS | 1.83 (0.99, 3.40) | 1.94 (0.99, 3.77) | 1.91 (0.93, 3.97) |
| **BMI[‡] (kg/m²)** | BMI 18.5 to ≤25.0 | | | |
| 25.0 to <30.0 | | **1.78 (1.02, 3.12)[‡]** | 1.58 (0.88, 2.84) | 0.93 (0.48, 1.81) |
| 30.0 to <35.0 | | **2.15 (1.17, 3.97)[‡]** | 1.81 (0.96, 3.40) | 0.73 (0.34, 1.59) |
| ≥35.0 | | 1.74 (0.92, 3.29) | 1.41 (0.69, 2.88) | 0.44 (0.18, 1.08) |
| **Youth weight[⁻]** | Normal weight | | | |
| A little overweight | | **2.69 (1.57, 4.62)[‡‡‡]** | – | **2.96 (1.58, 5.56)[‡‡‡]** |
| Moderately / very overweight | | **4.64 (2.51, 8.58)[‡‡‡]** | – | **6.14 (2.74, 13.72)[‡‡‡]** |

*Adjusted for age, country of residence, PCOS and BMI

[t]Adjusted for all in model 1 plus youth size category

[‡]Underweight category excluded due to inadequate sample size (n = 5)

[⁻]Youth weight, description of weight between 8 and 20 years of age. Underweight excluded due to inadequate sample size (n = 16). Moderately and very overweight categories combined (n = 76 for moderately overweight and n = 32 for very overweight)

[‡]$p < 0.05$

[‡‡]$p \leq 0.01$

[‡‡‡]$p \leq 0.001$

BMI, body mass index; PCOS, polycystic ovary syndrome

breast growth were 2.35 (95% CI: 1.16, 4.78) times higher as compared to women with pregnancy breast growth (adjusting for age, country of residence, GDM and BMI (S6 Table, Model 2).

## Discussion

To our knowledge, this is the first study reporting the proportion of anatomical breast characteristics among women from three countries who self-report insufficient milk supply. In our sample, 68% of women reported having at least one atypical breast (i.e., a type 2, 3 or 4 breast per S1 Fig in Huggins et al [25]). Over 80% of our sample responded to the breast anatomy survey questions demonstrating it is feasible to ask women to self-report markers suggestive of breast hypoplasia, and providing confidence about the content validity of our findings.

Lack of pregnancy breast growth, breast asymmetry and the presence of stretch marks on the breast have been identified as potential markers of breast hypoplasia [6]. Over three-quarters (76%) of women in our sample reported a lack of pregnancy breast growth. This figure is considerably higher than 24% (75/319) of healthy breastfeeding primiparous women and 18% (35/192) of postnatal women with a BMI <27 reporting this phenomenon [23, 39]. Also, in a socioeconomic diverse cohort of primiparous women, 7% (30/431) reported no prenatal breast enlargement [29]. The difference in these rates may be explained by our sample being women reporting low milk supply. Breast asymmetry was examined antenatally by Neifert et al who found that 8% (24/319) had 'moderate' and 0.3% (1/319) 'marked' asymmetry (no further detail is provided about these descriptions) [23]. Similarly, in our sample, 8% of women reported a ≥ 2 cup size difference between their breasts. As reported by Picard and colleagues, the breasts of 800 consecutive women (with a mean BMI of 23 and mean age of 26 years) were examined by the same dermatologist and the prevalence of breast stretch marks was 33% [40]. Obesity, higher pre-pregnancy BMI and higher gestational weight gain have been identified as risk factors for the development of stretch marks in pregnancy [40, 41]. In our sample, 72% of women reported the presence of stretch marks on their breasts prior to the birth of their first

child. Further research is needed about the timing of development and appearance of stretch marks in the general population.

Obesity is a significant public health concern in high and middle income countries with data showing the prevalence of obesity among reproductive age women to be over 40% in the USA and 30% in Australia and England [42–44]. Obesity is common among women self-reporting low milk supply and has been linked to decreased breastfeeding initiation, shorter breastfeeding duration, lower milk supply and delayed secretory activation [45, 46]. Vanky et al's study of 186 women with PCOS found those with no increase in bra size during pregnancy had larger BMIs compared with those who experienced breast size increment [11]. In our study, adjusted multiple logistic regression analyses revealed being overweight during pubertal years was strongly associated with having at least one atypical breast and lack of pregnancy breast growth, even after adjusting for pre-pregnancy BMI status. These novel results suggest that puberty is a sensitive window of mammary development and excess body fat during this time may be particularly impactful on lactation outcomes. In mice and rabbits, a high-fat or obesogenic diet during puberty increased the adiposity of the mammary glands and changed the shape of the alveoli in adulthood [10]. Similar findings have been found in research on Holstein heifers where high pre-pubertal growth rates have been linked to poorer mammary gland development (as determined by mammary DNA) [16, 47].

The outward appearance of breasts may or may not reflect insufficient glandular tissue. Balcar and colleagues used soft tissue radiography to examine the breasts of 61 women (mean age 23 years) with Stein-Leventhal syndrome (known today as polycystic ovary syndrome) [48]. These women's breasts were compared to 256 women without the condition [48]. Radiographs of the women's breasts revealed no clear relationship between the outward appearance of their breasts and the amount of glandular tissue [48].

We investigated whether a relationship exists between atypical breast type and other proposed markers of breast hypoplasia and found evidence of a link between atypical breast type and both lack of pregnancy breast growth and widely spaced breasts. This supports the findings by Huggins et al who found that among women with type 2, 3 or 4 breasts, 76% (22/29) and 86% (25/29) also had minimal or no pregnancy breast growth and widely spaced breasts respectively [25].

## Limitations

There are several limitations of this study. We used a convenience sample of women who were members of low milk supply online support groups. The sample was self-selected and biased to well-educated mothers, with a high breastfeeding intention. The reasons for the women's low milk supply are unknown. All exposure and outcome variables were identified via self-report and therefore lack objectivity, and we recognise that recall and confirmation biases are possible. The survey was accessed via Facebook, limiting access to women without the internet or social media accounts. Another limitation of our study is the use of BMI as a measure of adiposity, which is increasingly being recognised as insufficient as a single measure of metabolic health [49, 50]. We attempted to mitigate this limitation by inquiring about other measures of metabolic health such as diabetes, PCOS, and youth size.

The high proportion of women self-reporting low supply in this sample with various proposed breast hypoplasia markers cannot be determinative until compared to a reference population of women with normal milk production.

## Clinical and research implications

In addition to considering endocrine, obstetric, neonatal and social factors contributing to milk production, clinicians should be alert to consider breast hypoplasia as a possible diagnosis

when encountering women with a concern about milk supply [51]. Clinicians should ask breastfeeding women about gestational breast growth and early postpartum breast changes as well as examining their breast shape and intermammary width. To improve breastfeeding rates among larger women, targeted interventions which are equitable, accessible, relevant and non-stigmatising are required [52].

Future research is needed to examine reasons why women report low milk supply as the most common reason for ceasing breastfeeding prematurely. Projects are needed to increase knowledge about all the determinants of insufficient milk production, including breast hypoplasia despite the methodological challenges [53]. Research comparing the breast anatomy of women with low supply versus women who make a full supply would be useful to assist with determining which characteristics provide the strongest indication for the risk of low milk production. It would also be valuable to assess the relationship between deficit in maternal milk production (e.g. 75% deficit of daily volume [54] and breast hypoplasia markers.

## Conclusion

Members of Facebook groups for women with low milk supply have had high rates of atypical breasts (at least one breast being type 2, 3 or 4 as per S1 Fig in Huggins et al [25]), and often reported no breast growth in pregnancy. Women with larger bodies, and in particular, larger body weight during puberty, were more likely to have a number of features of breast hypoplasia including atypical breasts, widely spaced breasts, stretch marks on the breast and lack of pregnancy breast growth. To ascertain the strongest set of breast hypoplasia markers for predicting low supply, these findings must be confirmed in large well-designed cohort studies. Fundamental to helping more women to make a full milk supply to enable exclusive breastfeeding is an understanding that breastfeeding is a physiological function that promotes maternal physical and mental health [55]. When women encounter difficulty conceiving, they seek to understand why and treatment to help. Likewise, women unable to make a full milk supply also deserve to have their challenges investigated and explained. Therefore, it is time that human lactation became a research priority.

## Supporting information

**S1 Checklist. STROBE statement—checklist of items that should be included in reports of observational studies.**
(DOCX)

**S1 Fig. Breast types [25].**
(TIF)

**S2 Fig. Breast shape (adapted from Huggins et al [25]).**
(TIF)

**S1 Table. Facebook groups where invitations were posted.**
(PDF)

**S2 Table. Proportion of participants who answered questions about markers suggestive of breast hypoplasia.**
(PDF)

**S3 Table. Relationship between breast type category and delay in secretory activation.**
(PDF)

**S4 Table. Logistic regression modelling of risk factors for widely spaced breasts.**
(DOCX)

**S5 Table. Logistic regression modelling of risk factors for presence of stretch marks prior to birth of first child.**
(DOCX)

**S6 Table. Logistic regression modelling of risk factors for presence of lack of breast growth in pregnancy with first child.**
(DOCX)

**S1 File. Main survey.**
(PDF)

**S2 File. Eligibility check.**
(PDF)

**S3 File. Checklist for Reporting Results of Internet E-Surveys (CHERRIES).**
(PDF)

## Acknowledgments

We would like to thank all women who participated in this study and the Facebook administrators for allowing us to advertise our study in their groups. We acknowledge support from the National Institute of Child Health and Development grant number 1R01HD109915-01 (LNR). The authors are solely responsible for this manuscript's contents, findings, and conclusions, which do not necessarily represent the views of NIH.

## Author Contributions

**Conceptualization:** Renee L. Kam, Lisa H. Amir, Meabh Cullinane, Jenny Ingram, Laurie A. Nommsen-Rivers.

**Formal analysis:** Renee L. Kam, Xia Li, Laurie A. Nommsen-Rivers.

**Investigation:** Renee L. Kam.

**Methodology:** Renee L. Kam, Lisa H. Amir, Meabh Cullinane, Jenny Ingram, Xia Li, Laurie A. Nommsen-Rivers.

**Project administration:** Renee L. Kam.

**Supervision:** Lisa H. Amir, Meabh Cullinane.

**Writing – original draft:** Renee L. Kam.

**Writing – review & editing:** Renee L. Kam, Lisa H. Amir, Meabh Cullinane, Jenny Ingram, Xia Li, Laurie A. Nommsen-Rivers.

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
