## [Decision Letter · Decision Letter 0]

3 Oct 2023

PONE-D-23-19510Breast hypoplasia markers among women who report insufficient milk production: A retrospective online surveyPLOS ONE

Dear Dr. Kam,

Thank you for submitting your manuscript to PLOS ONE. After careful consideration, we feel that it has merit but does not fully meet PLOS ONE’s publication criteria as it currently stands. Therefore, we invite you to submit a revised version of the manuscript that addresses the points raised during the review process.

Please submit your revised manuscript by Nov 17 2023 11:59PM. If you will need more time than this to complete your revisions, please reply to this message or contact the journal office at plosone@plos.org. Please include the following items when submitting your revised manuscript:A rebuttal letter that responds to each point raised by the academic editor and reviewer(s). You should upload this letter as a separate file labeled 'Response to Reviewers'.A marked-up copy of your manuscript that highlights changes made to the original version. You should upload this as a separate file labeled 'Revised Manuscript with Track Changes'.An unmarked version of your revised paper without tracked changes. You should upload this as a separate file labeled 'Manuscript'.

We look forward to receiving your revised manuscript.

Kind regards,

Gilbert Sterling Octavius

Academic Editor

PLOS ONE

Journal Requirements:

5. Please upload a copy of Figure 1, to which you refer in your text on pages 8, 23 and 26. If the figure is no longer to be included as part of the submission please remove all reference to it within the text.

Reviewers' comments:

Reviewer's Responses to Questions

**Comments to the Author**

1. Is the manuscript technically sound, and do the data support the conclusions?

Reviewer #1: Yes

Reviewer #2: Partly

2. Has the statistical analysis been performed appropriately and rigorously? 

Reviewer #1: Yes

Reviewer #2: I Don't Know

3. Have the authors made all data underlying the findings in their manuscript fully available?

Reviewer #1: Yes

Reviewer #2: Yes

4. Is the manuscript presented in an intelligible fashion and written in standard English?

Reviewer #1: Yes

Reviewer #2: Yes

5. Review Comments to the Author

Reviewer #1: This study was well thought out, executed and analyzed, and the writing was largely clear and concise. The tables clearly laid out the factors and their various relationships the outcome and each other. The study adds to the progression of research on the topic. My comments relate largely to the background and discussion, which set the stage and then interpret the findings of the study. There were a few holes that I think filling in would add to the value of the paper. Otherwise, well done!

Notes:

Please define “index pregnancy” at the start. Some sources say it refers to the first pregnancy while other sources say it refers to the current baby. It was not until I reached line 169 that I finally knew for sure that the first baby was the index pregnancy.

Line 53: Geddes- was used to support the statement, ”When breast hypoplasia is present, there is a lack of sufficient glandular tissue [6].” However, Geddes did not actually study hypoplasia, but was rendering an educated opinion that low supply could be related to a possible deficiency of glandular tissue. What is missing is recognition of the fact that apparent hypoplasia of the breast- outward appearance—may or may not reflect actual hypoplasia of the glandular tissue, which is an important facet of this discussion, especially in light of the obesity angle. There were oblique hints to this fact later on but it was not explored directly. See this differentiation in Balcar, V., E. Silinkova-Malkova and Z. Matys (1972). "Soft tissue radiography of the female breast and pelvic pneumoperitoneum in the Stein-Leventhal syndrome." Acta Radiol Diagn (Stockh) 12(3): 353-362, which noted hypoplasia of the breast, hypoplasia of the gland, or both, as well as hypertrophic breasts with little glandular tissue (in dairy literature this is ‘fat heifer syndrome’). I would encourage the authors to acknowledge this differentiation even if it cannot be determined by the current study, because they are setting the stage for future research.

Line 57- It may be worth re-thinking the historical assumption that congenital causes of hypoplasia are due to syndromes or deformities. Some congenital hypoplasia may be due to fetal exposure to endocrine disruptors, altering the trajectory of eventual mammary development from birth. There is also a case report of missing estrogen receptors resulting in no mammary growth, likely congenital in nature, as well as a mitral valve prolapse connection that was noted in Rosenberg , C. A., Derman , G. H., Grabb , W. C., & Buda , A. J. (1983). Hypomastia and Mitral-Valve Prolapse. New England Journal of Medicine, 309(20), 1230-1232. doi:doi:10.1056/NEJM198311173092007 . See also tuberous breasts in next para.

Line 72: I think it is important to mention that Huggins based their drawings on the von Heimburg 1996 study and that their adaptation including making Type 1 the normal/reference breast, while in von Heimburg all 4 types had progressive deficiencies. Tuberous/tubal breasts are commonly discussed in the plastic surgery literature as a major form of hypoplasia and the original von Heimburg study was framed around progressive types of tuberous breast- Huggins does discuss this context. Wikipedia suggests that they are congenital, and don’t all fall under syndromes or chest wall deformities. Tuberous breasts - Wikipedia. It would be valuable to the reader to have some discussion of tuberous breasts woven into the discussion of hypoplasia.

Line 114- Was there a basis for using ≥ cup sizes as the criteria for breast asymmetry? Has this been defined/determined anywhere else?

Line 201- Who/what are these reference groups?

Line 206- So glad that the timing of onset of the conditions was included- this is tremendously important in the development of acquired mammary hypoplasia.

210: BMI- The authors are likely aware of the controversy surrounding the utility of standard BMI tools as applied to various ethnicities. This study had a small number of ethic (non-Caucasian?) participants, so the standard BMI may be appropriate here. However, it might be worth acknowledging this issue and commenting on the appropriateness of standard WHO BMI (which lumps everyone together) to your study respondents.

Line 221- were you referring to conditions that permanently alter lactation capacity only? Because each pregnancy/lactation cycle is a new opportunity for mammary growth or lack thereof, which can also be influenced by hormonal conditions or placental problems. Such problems can interfere with normal breasts reaching their potential in growth and be misconstrued as a permanent deficit.

319: It was mentioned on line 207 that data were collected about the timing of onset of endocrine conditions, but I don’t see mention of what was learned from this data, especially for onset of heaviness/obesity? This is important—see Hawkins, M. A. W., Colaizzi, J., Rhoades-Kerswill, S., Fry, E. D., Keirns, N. G., & Smith, C. E. (2019). Earlier Onset of Maternal Excess Adiposity Associated with Shorter Exclusive Breastfeeding Duration. J Hum Lact, 35(2), 292-300. doi:10.1177/0890334418799057

P 23 discussion: for future research, might I suggest adding the variable of percentage of milk produced to relate to the anatomical variables? Kuznetsov used the following for degrees of hypogalactia:

I - milk deficit is less than 25% of daily volume;

II - milk deficit is 26-50%;

III - milk deficit is 51-75%;

IV - milk deficit is greater than 75% [3].

Kuznetsov, V. (2017). Clinical and pathogenetic aspects of hypogalactia in post-parturient women. Актуальні проблеми сучасної медицини: Вісник української медичної стоматологічної академії, 17(1 (57)), 305-307.

P24 stretch mark discussion- I think it is important to look at whether the stretch marks are new (often red, at least in lighter skin) vs old (often silvery). Age/timing of development of these stretch marks (puberty? Pregnancy? Other?) may be significant; normally they are associated with windows of normal/rapid growth. Huggins notes on page 33 mentioned that many of their subjects reported they developed during adolescence but Huggins didn’t record this specifically. Collecting this info may provide more insight into the pathology of hypoplasia.

P 25-26- The statement is made that the findings “do not imply” that the markers are a risk factor for low supply. Page 27 then states that “to ascertain the strongest set of breast hypoplasia markers for predicting low supply….these findings must be confirmed in larger well-designed cohort studies” which does indeed seem to imply that the markers are risk factors. These conflict somewhat. Perhaps the first should be amended that these findings cannot be determinative until compared to a reference population. Since it is mentioned that these markers have not been tested in normal supply subjects, perhaps that needs to be part of the recommendation as well.

Reviewer #2: Abstract: Participants - information about recruitment from facebook groups is sketchy. For instance, number of online groups accessed and exclusions?

Introduction: More information about perceived low milk supply is needed. The authors appear to equate perceived low milk supply with confirmed low milk supply, without discussing that they may not align. More information is desirable on how the authors confirmed the mothers' impressions. Did they establish that the mothers were offering the breast unrestrictedly, for instance?

- The classification of the breast appearance uses an appropriate system, that of Huggins et al., the best that is currently in the literature.

- Acquired breast hypoplasia: Another cause has been omitted from mention, that is, as a consequence of breast reduction surgery, especially if the reduction was substantial. This consequence, as I have seen, can vary in different geographical settings, which perhaps the authors might want to investigate in a future paper..

Ethnicity: There are different criteria for ethnicity across the three national settings used by the authors.

Design: The authors have, rightly, mentioned the limitation regarding face-to-face research due to the COVID-19 pandemic.

The topic is a worthy one, but the above-mentioned flaws detract from the article.

6. PLOS authors have the option to publish the peer review history of their article (what does this mean?). If published, this will include your full peer review and any attached files.

Reviewer #1: No

Reviewer #2: **Yes: **Virginia Thorley

---

## [Author Response · Author response to Decision Letter 0]

14 Dec 2023

December 11th 2023

Re: Revisions to Manuscript ID PONE-D-23-19510

Dear Gilbert Sterling Octavius,

Many thanks to you and the reviewers for their assessment of our manuscript entitled "Breast hypoplasia markers among women who report insufficient milk production: A retrospective online survey”. We have addressed each of the required revisions and uploaded two versions of the revised manuscript – one with and one without tracked changes. 

Kind regards,

Renee Kam and co-authors

Editor comments:

Comment: 1. Please ensure that your manuscript meets PLOS ONE's style requirements, including those for file naming. The PLOS ONE style templates can be found at 

Response: Thank you, these links have been reviewed and changes made were needed. 

Comment: 2. Please provide additional details regarding participant consent. In the ethics statement in the Methods and online submission information, please ensure that you have specified what type you obtained (for instance, written or verbal, and if verbal, how it was documented and witnessed). If your study included minors, state whether you obtained consent from parents or guardians. If the need for consent was waived by the ethics committee, please include this information.

Response: Under the heading ‘Survey’, we say: 

“Women were screened using an eligibility survey (S2 File) where they were informed about the purpose of the study. Eligible participants could download and read the Participant Information Statement and were asked “Do you agree to complete the survey? Clicking 'Yes' tells us you want to take part in the study.” If a participant provided consent to complete the survey by clicking ‘Yes’, they were lead to the survey.” P10

Please advise if this is insufficient.

Comment: 3. We note that you have indicated that data from this study are available upon request. PLOS only allows data to be available upon request if there are legal or ethical restrictions on sharing data publicly. For more information on unacceptable data access restrictions, please see http://journals.plos.org/plosone/s/data-availability#loc-unacceptable-data-access-restrictions. 

Response: In our ethics application we stated that our data would not be “freely available to use, reuse and redistribute” and that we would not “make data publicly accessible”. Therefore, we cannot share the dataset openly. In the journal submission process regarding data availability we stated “Data cannot be shared publicly because we do not have approval from the La Trobe University Human Research Ethics Committee. The dataset is available from the corresponding author upon reasonable request”. 

Comment: 4. We note that you have included the phrase “data not shown” in your manuscript. Unfortunately, this does not meet our data sharing requirements. PLOS does not permit references to inaccessible data. We require that authors provide all relevant data within the paper, Supporting Information files, or in an acceptable, public repository. Please add a citation to support this phrase or upload the data that corresponds with these findings to a stable repository (such as Figshare or Dryad) and provide and URLs, DOIs, or accession numbers that may be used to access these data. Or, if the data are not a core part of the research being presented in your study, we ask that you remove the phrase that refers to these data.

Response: We noted there are three places where we say “data not shown”. For the first mention of this phrase, we have included the data in an additional supplementary file (S3 table). For the subsequent two mentions of this phrase, the text fully describes all the data and the tables do not show anything that the text does not. Therefore we have deleted the phrase in these instances. 

Comment: 5. Please upload a copy of Figure 1, to which you refer in your text on pages 8, 23 and 26. If the figure is no longer to be included as part of the submission please remove all reference to it within the text.

Response: We have deleted the first mention of this figure as it refers to Fig 1 in our report of the reliability study (Kam et al. 2021) and we think it not necessary to mention here. We mention ‘Fig 1’ for the second and third times in relation to the Fig 1 in the Huggins 2000 paper. We believe it is helpful to readers to indicate where these images originate.

Reviewer 1 comments:

Comment: This study was well thought out, executed and analyzed, and the writing was largely clear and concise. The tables clearly laid out the factors and their various relationships the outcome and each other. The study adds to the progression of research on the topic. My comments relate largely to the background and discussion, which set the stage and then interpret the findings of the study. There were a few holes that I think filling in would add to the value of the paper. Otherwise, well done!

Response: Thank you

Comment: Please define “index pregnancy” at the start. Some sources say it refers to the first pregnancy while other sources say it refers to the current baby. It was not until I reached line 169 that I finally knew for sure that the first baby was the index pregnancy.

Response: Sorry this wasn’t clear. We have changed all mentions of ‘index’ to ‘first’. 

Comment: Line 53: Geddes- was used to support the statement, ”When breast hypoplasia is present, there is a lack of sufficient glandular tissue [6].” However, Geddes did not actually study hypoplasia, but was rendering an educated opinion that low supply could be related to a possible deficiency of glandular tissue. What is missing is recognition of the fact that apparent hypoplasia of the breast- outward appearance—may or may not reflect actual hypoplasia of the glandular tissue, which is an important facet of this discussion, especially in light of the obesity angle. There were oblique hints to this fact later on but it was not explored directly. See this differentiation in Balcar, V., E. Silinkova-Malkova and Z. Matys (1972). "Soft tissue radiography of the female breast and pelvic pneumoperitoneum in the Stein-Leventhal syndrome." Acta Radiol Diagn (Stockh) 12(3): 353-362, which noted hypoplasia of the breast, hypoplasia of the gland, or both, as well as hypertrophic breasts with little glandular tissue (in dairy literature this is ‘fat heifer syndrome’). I would encourage the authors to acknowledge this differentiation even if it cannot be determined by the current study, because they are setting the stage for future research.

Response: We have changed the old sentence: “When breast hypoplasia is present, there is a lack of sufficient glandular tissue” to a new sentence: “Although the outward appearance of the breast is not conclusive, it is thought that breasts appearing as hypoplastic lack sufficient glandular tissue". P4

In addition, the following sentence has been added to the Discussion section: 

“The outward appearance of breasts may or may not reflect insufficient glandular tissue. Balcar and colleagues used soft tissue radiography to examine the breasts of 61 women (mean age 23 years) with Stein-Leventhal syndrome (known today as polycystic ovary syndrome) (48). These women’s breasts were compared to 256 women without the condition (48). Radiographs of the women’s breasts revealed no clear relationship between the outward appearance of their breasts and the amount of glandular tissue (48).” P30 in tracked version of manuscript and P27 in untracked version.

Comment: Line 57- It may be worth re-thinking the historical assumption that congenital causes of hypoplasia are due to syndromes or deformities. Some congenital hypoplasia may be due to fetal exposure to endocrine disruptors, altering the trajectory of eventual mammary development from birth. There is also a case report of missing estrogen receptors resulting in no mammary growth, likely congenital in nature, as well as a mitral valve prolapse connection that was noted in Rosenberg , C. A., Derman , G. H., Grabb , W. C., & Buda , A. J. (1983). Hypomastia and Mitral-Valve Prolapse. New England Journal of Medicine, 309(20), 1230-1232. doi:doi:10.1056/NEJM198311173092007 . See also tuberous breasts in next para.

Response: The sentence has been updated with the addition of the text in red below:

“Congenital breast hypoplasia is associated with uncommon syndromes (e.g. Poland or Jeune), chest wall deformities (e.g. pectus excavatum), mitral valve prolapse, or hormonal disruption due to oestrogen insensitivity (7-9). In addition, there is increasing concern that exposure to environmental contaminants in utero or during puberty may impair mammary gland development (10). P4

The case report of missing estrogen receptors resulting in no mammary growth (Quaynor et al 2013) has not been cited due to a lack of data pertaining to this cause. 

Comment: Line 72: I think it is important to mention that Huggins based their drawings on the von Heimburg 1996 study and that their adaptation including making Type 1 the normal/reference breast, while in von Heimburg all 4 types had progressive deficiencies. Tuberous/tubal breasts are commonly discussed in the plastic surgery literature as a major form of hypoplasia and the original von Heimburg study was framed around progressive types of tuberous breast- Huggins does discuss this context. Wikipedia suggests that they are congenital, and don’t all fall under syndromes or chest wall deformities. Tuberous breasts - Wikipedia. It would be valuable to the reader to have some discussion of tuberous breasts woven into the discussion of hypoplasia.

Response: Thank you for the suggestion. Additional text has been added: 

“Breast hypoplasia has been recognised as a characteristic of the ‘tuberous’ breast deformity in the plastic surgery literature since 1976 (24). Prior to our research, the largest study to investigate a possible relationship between anatomical breast characteristics suggestive of breast hypoplasia and milk production was a case series of 34 women conducted by Huggins et al (25) These researchers adapted a tool from a retrospective analysis of 40 patients undergoing operative breast corrections (26) to categorise women's breasts into one of four progressive types of tuberous breast (S1 Fig) (25). In Huggins and colleagues’ sample, the women’s ‘breast type’ appeared to be related to the adequacy of their milk production, as women with type 2, 3 or 4 breasts produced insufficient milk compared to women with type 1 (“typical appearance”) breasts (25).” P5

Comment: Line 114- Was there a basis for using ≥ cup sizes as the criteria for breast asymmetry? Has this been defined/determined anywhere else?

Response: Previous research by Huggins 2000 and Neifert 1990 reported that breast asymmetry was associated with breast hypoplasia, but used only descriptive terms to describe asymmetry. We wanted to describe the feature in more measurable terms. We selected ≥ 2 breast sizes because we could not identify an agreed measure of breast asymmetry.

Comment: Line 201- Who/what are these reference groups?

Response: The reference groups refer to the normal weight category for BMI or the absence of an endocrine disorder (eg PCOS, GDM). We have included the references groups in brackets within the text now to make this clearer. P11

Comment: Line 206- So glad that the timing of onset of the conditions was included- this is tremendously important in the development of acquired mammary hypoplasia.

Response: Thank you

Comment: 210: BMI- The authors are likely aware of the controversy surrounding the utility of standard BMI tools as applied to various ethnicities. This study had a small number of ethic (non-Caucasian?) participants, so the standard BMI may be appropriate here. However, it might be worth acknowledging this issue and commenting on the appropriateness of standard WHO BMI (which lumps everyone together) to your study respondents.

Response: We have added the following sentence in the DISCUSSION section: 

“Another limitation of our study is the use of BMI as a measure of adiposity, which is increasingly being recognised as insufficient as a single measure of metabolic health (49, 50). We attempted to mitigate this limitation by inquiring about other measures of metabolic health such as diabetes, PCOS, and youth size.” P31 in tracked version of manuscript and P28 in untracked version.

Comment: Line 221- were you referring to conditions that permanently alter lactation capacity only? Because each pregnancy/lactation cycle is a new opportunity for mammary growth or lack thereof, which can also be influenced by hormonal conditions or placental problems. Such problems can interfere with normal breasts reaching their potential in growth and be misconstrued as a permanent deficit.

Response: We can see how the following sentence in the previous version of the manuscript would cause confusion: 

Previous version: “'Participants were also asked about conditions, obstetric history (method of birth and analgesia used during labour) or surgery which may interfere with mammary glandular tissue development and/or lactation capacity.” 

Therefore, we have updated the sentence in the revised manuscript as follows:

Revised version: “Participants were also asked about medical conditions and obstetric history as these covariates may influence lactation outcomes.” P13 in tracked version of manuscript and P12 in untracked version.

Comment: 319: It was mentioned on line 207 that data were collected about the timing of onset of endocrine conditions, but I don’t see mention of what was learned from this data, especially for onset of heaviness/obesity? This is important—see Hawkins, M. A. W., Colaizzi, J., Rhoades-Kerswill, S., Fry, E. D., Keirns, N. G., & Smith, C. E. (2019). Earlier Onset of Maternal Excess Adiposity Associated with Shorter Exclusive Breastfeeding Duration. J Hum Lact, 35(2), 292-300. doi:10.1177/0890334418799057

Response: Thank you for this feedback. We have now undertaken additional analyses to assess the relationship between women’s description of their weight between 8 and 20 years of age (‘youth weight’) and the various breast anatomy characteristics. The results of these analyses have been added to tables 4 and 5. 

Within the revised manuscript we have added the following text corresponding to our further analyses including the youth weight variable:

ABSTRACT: Multiple logistic regression analyses identified overweight during pubertal years as a risk factor for atypical breast type and lack of pregnancy breast growth. 

Participants in low milk supply Facebook groups reported high rates of breast hypoplasia markers. Overweight during adolescence is a risk factor for breast hypoplasia markers.

METHODS section: Participants were asked to describe their weight between 8 and 20 years of age using the following categories: ‘underweight’, ‘normal weight’, ‘a little overweight’, ‘moderately overweight’, ‘very overweight’, ‘unsure’, ‘prefer not to say’ or ‘other’. We refer to this variable as ‘youth weight’. P12

RESULTS section: Using Chi-square analyses, we explored bivariate relationships between suggested markers of breast hypoplasia, BMI, youth weight, PCOS, GDM and hypothyroidism and the presence of at least one atypical breast (Table 4). Women with a high BMI (≥25 kg/m2) were more likely to report atypical compared to typical breasts (Cramer’s V=0.1487 [small effect size] (1), p=0.036). Women who described being overweight between 8 and 20 years of age were more likely to report atypical compared to typical breasts (Cramer’s V=0.2875 [medium effect size], p<0.001). P19

RESULTS section: Using Chi-square analyses, we explored relationships between endocrine conditions (including BMI and youth weight) and suggested markers of breast hypoplasia (Table 5). Women with a high BMI were more likely to have widely spaced breasts, stretch marks present on their breasts and lack of pregnancy breast growth (Cramer’s V=0.1870 [small effect size] , p=0.002; Cramer’s V=0.2150 [small effect size], p=0.01; Cramer’s V=0.2250 [small effect size], p<0.001 respectively). Women who described being overweight between 8 and 20 years of age were more likely to have widely spaced breasts and a lack of pregnancy breast growth (Cramer’s V=0.1867 [small effect size], p=0.001; Cramer’s V=2123 [small effect size], p<0.001) respectively). P21 in tracked version of manuscript and P20 of untracked version.

RESULTS section: We performed regression analyses on the four outcomes for which Chi-square analyses had at least one predictor variable with p-value <0.10 (see Tables 4 and 5). Crude and adjusted odds ratios were obtained for these relationships by performing logistic regression analyses in stages. In the first stage of adjusted analyses, we adjusted for significant socio-demographic variables and current metabolic health variables. In the second stage we added in youth weight status to determine its direct effect on each outcome independent of current metabolic health variables. P24 in tracked version of manuscript and P23 of untracked version.

RESULTS section: Although BMI category was a significant predictor of atypical breasts in the unadjusted analysis, it was no longer significant after adjusting for age, country of residence, and PCOS status (Table 6, Model 1). However, youth weight category remained a strong predictor of atypical breasts in a model adjusted for all of these covariates (Table 6, Model 2). The odds of having at least one atypical breast were 2.96 (95% CI: 1.58, 5.56) and 6.14 (95% CI: 2.74, 13.72) times higher among women who described being a ‘little overweight’ and ‘moderately or very overweight’ between 8 and 20 years of age, respectively, compared to women without at least one atypical breast (adjusting for age, country of residence, PCOS status and BMI) (Table 6, Model 2). P24-25 in tracked version of manuscript and P23 in tracked version.

RESULTS section: Logistic regression modelling between various metabolic health exposures and widely spaced breasts, stretch marks on the breast and lack of pregnancy breast growth were also undertaken (S4-6 Tables). BMI category was a significant predictor of widely spaced breasts, stretch marks on the breast and lack of pregnancy growth in the unadjusted analyses (S4 Table, Crude OR). However, it only remained a significant predictor for widely spaced breasts in the obese 1 category after adjusting for country of residence and USA ethnicity (S4 Table, Model 1). Also, BMI was no longer a significant predictor for stretch marks on the breast after adjusting for USA ethnicity and PCOS status (S5 Table, Model 1). P26 in tracked version of manuscript and P24 in untracked version.

Although BMI category was a significant predictor of lack of pregnancy breast growth in the unadjusted analysis, it was no longer significant after adjusting for age, country of residence, and GDM status (S6 Table, Model 1). However, the moderately / very overweight youth weight category remained a strong predictor of lack of pregnancy breast growth in a model adjusted for all of these covariates (S6 Table, Model 2). Among women who described being ‘moderately or very overweight’ between 8 and 20 years of age, the odds of lack of pregnancy breast growth were 2.35 (95% CI: 1.16, 4.78) times higher as compared to women with pregnancy breast growth (adjusting for age, country of residence, GDM and BMI (S6 Table, Model 2). P27 of tracked version of manuscript and P25 of untracked version.

DISCUSSION section: Obesity is a significant public health concern in high and middle income countries with data showing the prevalence of obesity among reproductive age women to be over 40% in the USA and 30% in Australia and England (42-44). Obesity is common among women self-reporting low milk supply and has been linked to decreased breastfeeding initiation, shorter breastfeeding duration, lower milk supply and delayed secretory activation (45, 46). Vanky et al’s study of 186 women with PCOS found those with no increase in bra size during pregnancy had larger BMIs compared with those who experienced breast size increment (11). In our study, adjusted multiple logistic regression analyses revealed being overweight during pubertal years was strongly associated with having at least one atypical breast and lack of pregnancy breast growth, even after adjusting for pre-pregnancy BMI status. These novel results suggest that puberty is a sensitive window of mammary development and excess body fat during this time may be particularly impactful on lactation outcomes. In mice and rabbits, a high-fat or obesogenic diet during puberty increased the adiposity of the mammary glands and changed the shape of the alveoli in adulthood (10). Similar findings have been found in research on Holstein heifers where high pre-pubertal growth rates have been linked to poorer mammary gland development (as determined by mammary DNA) (16, 47). P29 in tracked version of manuscript and P27 in untracked version.

Comment: P 23 discussion: for future research, might I suggest adding the variable of percentage of milk produced to relate to the anatomical variables? Kuznetsov used the following for degrees of hypogalactia:

I - milk deficit is less than 25% of daily volume;

II - milk deficit is 26-50%;

III - milk deficit is 51-75%;

IV - milk deficit is greater than 75% [3].

Kuznetsov, V. (2017). Clinical and pathogenetic aspects of hypogalactia in post-parturient women. Актуальні проблеми сучасної медицини: Вісник української медичної стоматологічної академії, 17(1 (57)), 305-307.

Response: Thank you for this suggestion. We would like to add the following new sentence at the end of the Discussion “It would also be valuable to assess the relationship between deficit in maternal milk production (e.g. <25% / 26-50% /51-75% / >75% deficit of daily volume [ref]) and breast hypoplasia markers.” However, we unfortunately have not been able to source the journal article and hence cannot cite it. We are therefore wondering if you could help us source it? 

Comment: P24 stretch mark discussion- I think it is important to look at whether the stretch marks are new (often red, at least in lighter skin) vs old (often silvery). Age/timing of development of these stretch marks (puberty? Pregnancy? Other?) may be significant; normally they are associated with windows of normal/rapid growth. Huggins notes on page 33 mentioned that many of their subjects reported they developed during adolescence but Huggins didn’t record this specifically. Collecting this info may provide more insight into the pathology of hypoplasia.

Response: We did collect data about the timing of onset of stretch marks. See Table 3. 

We already say in the Discussion: “As reported by Picard and colleagues, the breasts of 800 consecutive women (with a mean BMI of 23 and mean age of 26 years) were examined by the same dermatologist and the prevalence of breast stretch marks was 33% (40). Obesity, higher pre-pregnancy BMI and higher gestational weight gain have been identified as risk factors for the development of stretch marks in pregnancy (40, 41). In our sample, 72% of women reported the presence of stretch marks on their breasts prior to the birth of their first child.” 

After this, we have added the following sentence: 

“Further research is needed about the timing of development and appearance of stretch marks in the general population.” P28-29 in tracked version of manuscript and P26 in untracked version.

Comment: P 25-26- The statement is made that the findings “do not imply” that the markers are a risk factor for low supply. Page 27 then states that “to ascertain the strongest set of breast hypoplasia markers for predicting low supply….these findings must be confirmed in larger well-designed cohort studies” which does indeed seem to imply that the markers are risk factors. These conflict somewhat. Perhaps the first should be amended that these findings cannot be determinative until compared to a reference population. Since it is mentioned that these markers have not been tested in normal supply subjects, perhaps that needs to be part of the recommendation as well.

Response: The first sentence in the Discussion mentioned above has been edited. 

Previous version:

“The high proportion of women in this sample with various proposed breast hypoplasia markers does not imply that these factors are ‘risks’ for low milk supply since the proportion of women with normal milk production/general population with these markers is unknown.”:

Revised version:

“The high proportion of women self-reporting low supply in this sample with various proposed breast hypoplasia markers cannot be determinative until compared to a reference population of women with normal milk production/general population.” P31 of tracked version of manuscript and P28 of untracked version.

Reviewer 2 comments:

Comment: Abstract: Participants - information about recruitment from facebook groups is sketchy. For instance, number of online groups accessed and exclusions?

Response: We have included the number of Facebook groups recruited from in the Abstract now. 

“Setting: Five low milk supply Facebook groups.”

The exclusion criteria are covered under the heading ‘Participants’ in the Abstract. 

Comment: Introduction: More information about perceived low milk supply is needed. The authors appear to equate perceived low milk supply with confirmed low milk supply, without discussing that they may not align. More information is desirable on how the authors confirmed the mothers' impressions. Did they establish that the mothers were offering the breast unrestrictedly, for instance?

Response: We agree it can be hard to determine true low milk supply. We have added the following text about perceived low milk supply in the Introduction: 

“The proportion of women ceasing breastfeeding early who have a perceived versus an actual insufficient milk production is unknown. A perceived low supply occurs when a mother believes her supply is insufficient regardless of whether an actual low supply exists or not (4). An actual low supply can result from breastfeeding challenges (secondary low supply) or can inherently exist (primary low supply) (5).” P4

As a part of our screening for eligibility for the project, we asked women if they were removing milk at least 6 times per 24 hours prior to assuming they had low milk supply. If mother-infant dyads were separated or mothers not expressing/feeding at least six times per 24 hours there were ineligible.

Comment: - The classification of the breast appearance uses an appropriate system, that of Huggins et al., the best that is currently in the literature.

Response: Thank you

Comment: - Acquired breast hypoplasia: Another cause has been omitted from mention, that is, as a consequence of breast reduction surgery, especially if the reduction was substantial. This consequence, as I have seen, can vary in different geographical settings, which perhaps the authors might want to investigate in a future paper.

Response: We have included breast reduction surgery as a reason for acquired breast hypoplasia. 

“Acquired breast hypoplasia can be associated with a history of breast radiation, breast reduction surgery or breast haemangioma (7).” P4

Comment: Ethnicity: There are different criteria for ethnicity across the three national settings used by the authors.

Response: Yes, that’s correct. We have authors from each national setting where women were recruited from and standard categories for ethnicity were chosen to best reflect each setting.

Comment: Design: The authors have, rightly, mentioned the limitation regarding face-to-face research due to the COVID-19 pandemic.

Response: Thank you

Other:

1. We identified that we referred to supplementary files inaccurately within the text of the

previous manuscript and therefore have rectified these in the revised manuscript as follows:

‘(Fig in S1 Fig)’ replaced with ‘(S1 Fig)’ P5 & 7

‘(Table in S1 Table)’ repaced with ‘(S1 Table)’ P9 

‘(File in S1 File)’ replaced with ‘(S1 File)’ P10

‘(File in S2 File)’ replaced with ‘(S2 File)’ P10 

‘Figure in S2 Fig’ replaced with ‘S2 Fig’ P11

‘(File in S3 File)’ replaced with ‘(S3 File)’ P14

‘(Table in S2 Table)’ replaced with ‘(S2 Table)’ P17

2. We have edited the following paragraph from the METHODS section and moved it to the DISCUSSION section as follows:

Previous version: ‘We acknowledge the controversy surrounding the utility of standard BMI tools being used as a single measure (2), however we felt the use of this tool was appropriate for our study given the majority (>90%) of participants identify as white.’

Revised version: ‘Another limitation of our study is the use of BMI as a measure of adiposity, which is increasingly being recognised as insufficient as a single measure of metabolic health (49, 50). We attempted to mitigate this limitation by inquiring about other measures of metabolic health such as diabetes, PCOS, and youth size.’ P31 in tracked version of manuscript and P28 of untracked version.

3. We identified in the RESULTS section that we did not describe the comparison group adequately for one of the analyses and hence have rectified this as follows:

Previous version: Women who reported a delay or absence in secretory activation were more likely to report at least one atypical breast (71%; 216/304) compared to other women (29%; 88/304) (χ2(1) 4.1122, p=0.043) 

Revised version: Women who reported a delay or absence in secretory activation were more likely to report at least one atypical breast (71%; 216/304) compared to women without at least one atypical breast (29%; 88/304) (χ2(1) 4.1122, p=0.043) P24 of tracked version of manuscript and P23 of untracked version.

4. When undertaking the additional analyses with the ‘youth weight’ variable included, we 

identified that our previous multiple logistic regression analyses had not included all covariates from Table 1 and metabolic health exposures (including BMI) for which bivariate analyses revealed a p value <0.1. In addition, we identified that in our previous multiple logistic regression analyses that we included some covariates from Table 1 metabolic health exposures (including BMI) which had not revealed a p value <0.1. Therefore, we conducted the logistic regression analyses again including covariates from Table 1 and exposures where a p value < 0.1 was revealed from bivariate analyses. 

To make it clear which sociodemographic characteristics displayed in table 1 and breast characteristics chi-square analyses revealed associations p<0.1, we included the following paragraph:

“Chi-square analyses between sociodemographic characteristics displayed in table 1 and breast characteristics which demonstrated associations where p<0.10 included breast type and i) age (χ2(1)=6.2999, p=0.012) and ii) country of residence (χ2(1)=8.3689, p=0.004); widely spaced breasts and i) country of residence (χ2(1)=7.6443, p=0.006) and ii) USA ethnicity (χ2(1)=2.7848, p=0.095); asymmetry and UK ethnicity (χ2(1)=9.4138, p=0.002); lack of growth in pregnancy with first child and i) age (χ2(1)=3.15078, p=0.076) and ii) country of residence (χ2(1)=2.7586, p=0.097); presence of stretch marks prior to birth of first child and USA ethnicity (χ2(1)=4.4058, p=0.036).” P16 in tracked version of manuscript and P15-16 in untracked version.

5. We decided that the Chi square analyses assessing a relationship between atypical breast type and other proposed breast hypoplasia markers was sufficient and that multiple logistic regression was not necessary. Therefore, we have removed the following paragraph at the end of the RESULTS section:

‘We used multiple logistic regression analyses to investigate whether women with atypical breasts might be more likely to have other proposed breast hypoplasia markers. When adjusted for BMI, PCOS, GDM, hypothyroidism, age and country of residence, the odds of women having at least one atypical breast was 8.86 times higher in women with widely spaced breasts and 3.39 times higher in women with a lack of pregnancy breast growth compared to women without at least one atypical breast (95% CI: 4.88, 16.07; 95% CI: 2.01, 5.71) (data not shown).’ 

6. Reviewer 1’s feedback resulted in us identifying the ‘youth weight’ variable as an important predictor variable for breast characteristics. This prompted us to review how to present the logistic regression data. We decided the most robust approach for each logistic regression table to have 3 levels of modelling as follows:

a. Column 1 displays the crude odds ratio for each covariate/exposure variable selected for inclusion in multiple variable logistic regression. 

b. Column 2 displays the adjusted odds, adjusting for all ‘qualifying covariates/exposures EXCEPT youth size category.

c. Column 3 includes the same variables as in column 2, plus youth size category

Based on points 3 and 4 above, we have edited the logistic regression results in the RESULTS section as follows:

Previous version: We performed nine separate multiple logistic regression analyses on the relationships for which Chi-square analyses had a p value <0.1. These relationships included between: i) BMI and atypical breast type, ii) BMI and widely spaced breasts, iii) BMI and stretch marks on the breast, iv) BMI and lack of pregnancy breast growth; v) PCOS and atypical breast type, vi) PCOS and stretch marks on the breast, vii) widely spaced breasts and atypical breast type, viii) lack of pregnancy breast growth and atypical breast type, and ix) GDM and lack of pregnancy breast growth. Crude and adjusted odds ratios were obtained for these relationships by performing bivariate logistic regression analyses and multiple logistic regression analyses adjusting for covariates. 

Various relationships between BMI and proposed markers of breast hypoplasia remained significant in multiple logistic regression models after adjusting for covariates (age, country of residence, PCOS, GDM and hypothyroidism) (Table in S3 Table). The normal weight category was used as the reference category. In the adjusted model, the odds of having widely spaced breasts were 1.97 (95% CI: 1.17, 3.32) times and 2.21 (95% CI: 1.26, 3.86) higher among women in the overweight and obese 1 category, respectively, compared to women with normal weight. The odds of having stretch marks on the breast were 3.97 (95% CI: 1.55, 10.14) times higher among women in the obese 1 category compared to women with normal weight. The odds of a lack of pregnancy breast growth were also significantly more likely among women in the overweight (2.27 [95% CI: 1.29, 4.02]), obese 1 (2.07 [95% CI: 1.12, 3.80]) and obese 2+ (3.52 [95% CI: 1.63, 7.58]) categories. The relationship between BMI and atypical breast type was no longer significant in the adjusted model. 

When adjusted for BMI, country of residence and age, multiple logistic regression analyses revealed no evidence of an association between PCOS and the presence of stretch marks on the breast (1.79 [95% CI: 0.72, 4.42]) nor between GDM and lack of pregnancy growth (1.61 [95% CI: 0.74, 3.54]) and some evidence of a relationship between PCOS and atypical breast type (1.94 [95% CI: 0.99, 3.77]) (data not shown). 

We used multiple logistic regression analyses to investigate whether women with atypical breasts might be more likely to have other proposed breast hypoplasia markers. When adjusted for BMI, PCOS, GDM, hypothyroidism, age and country of residence, the odds of women having at least one atypical breast was 8.86 times higher in women with widely spaced breasts and 3.39 times higher in women with a lack of pregnancy breast growth compared to women without at least one atypical breast (95% CI: 4.88, 16.07; 95% CI: 2.01, 5.71) (data not shown).

Revised version: We performed regression analyses on the four outcomes for which Chi-square analyses had at least one predictor variable with p-value <0.10 (see Tables 4 and 5). Crude and adjusted odds ratios were obtained for these relationships by performing logistic regression analyses in stages. In the first stage of adjusted analyses, we adjusted for significant socio-demographic variables and current metabolic health variables. In the second stage we added in youth weight status to determine its direct effect on each outcome independent of current metabolic health variables.

Although BMI category was a significant predictor of atypical breasts in the unadjusted analysis, it was no longer significant after adjusting for age, country of residence, and PCOS status (Table 6, Model 1). However, youth weight category remained a strong predictor of atypical breasts in a model adjusted for all of these covariates (Table 6, Model 2). The odds of having at least one atypical breast were 2.96 (95% CI: 1.58, 5.56) and 6.14 (95% CI: 2.74, 13.72) times higher among women who described being a ‘little overweight’ and ‘moderately or very overweight’ between 8 and 20 years of age, respectively, compared to women without at least one atypical breast (adjusting for age, country of residence, PCOS status and BMI) (Table 6, Model 2).

Logistic regression modelling between various metabolic health exposures and widely spaced breasts, stretch marks on the breast and lack of pregnancy breast growth were also undertaken (S4-6 Tables). BMI category was a significant predictor of widely spaced breasts, stretch marks on the breast and lack of pregnancy growth in the unadjusted analyses (S4 Table, Crude OR). However, it only remained a significant predictor for widely spaced breasts in the obese 1 category after adjusting for country of residence and USA ethnicity (S4 Table, Model 1). Also, BMI was no longer a significant predictor for stretch marks on the breast after adjusting for USA ethnicity and PCOS status (S5 Table, Model 1). 

Although BMI category was a significant predictor of lack of pregnancy breast growth in the unadjusted analysis, it was no longer significant after adjusting for age, country of residence, and GDM status (S6 Table, Model 1). However, the moderately / very overweight youth weight category remained a strong predictor of lack of pregnancy breast growth in a model adjusted for all of these covariates (S6 Table, Model 2). Among women who described being ‘moderately or very overweight’ between 8 and 20 years of age, the odds of lack of pregnancy breast growth were 2.35 (95% CI: 1.16, 4.78) times higher as compared to women with pregnancy breast growth (adjusting for age, country of residence, GDM and BMI (S6 Table, Model 2).

We used multiple logistic regression analyses to investigate whether women with atypical breasts might be more likely to have other proposed breast hypoplasia markers. When adjusted for BMI, youth weight, country of residence and USA ethnicity, the odds of women having at least one atypical breast was 7.01 times higher in women with widely spaced breasts compared to women without at least one atypical breast (95% CI: 3.42, 14.34). Adjusting for GDM, BMI, youth weight, age and country of residence, the odds of women having at least one atypical breast was 2.68 times higher in women with a lack of pregnancy breast growth compared to women without one atypical breast 95% CI: 1.56, 4.62). 

7. We have edited the following paragraph in the DISCUSSION section as follows to make it clearer:

Previous version: ‘This figure is considerably higher than 24% (75/319) and 18% (35/192) reporting this phenomenon among healthy breastfeeding primiparous women and primiparous (42%) and multiparous (28%) women with a BMI <27, respectively, who gave birth to healthy term newborns (2, 3).’

Revised version: ‘This figure is considerably higher than 24% (75/319) of healthy breastfeeding primiparous women and 18% (35/192) of postnatal women with a BMI <27 reporting this phenomenon (23, 39).’ P28 of tracked version of manuscript and P25-56 of untracked version.

---

## [Decision Letter · Decision Letter 1]

22 Jan 2024

PONE-D-23-19510R1Breast hypoplasia markers among women who report insufficient milk production: A retrospective online surveyPLOS ONE

Dear Dr. Renee,

Thank you for submitting your manuscript to PLOS ONE. After careful consideration, we feel that it has merit but does not fully meet PLOS ONE’s publication criteria as it currently stands. Therefore, we invite you to submit a revised version of the manuscript that addresses the points raised during the review process.

**ACADEMIC EDITOR: **

Please address the concerns of the reviewers

We look forward to receiving your revised manuscript.

Kind regards,

Gilbert Sterling Octavius

Academic Editor

PLOS ONE

Journal Requirements:

Reviewers' comments:

Reviewer's Responses to Questions

**Comments to the Author**

1. If the authors have adequately addressed your comments raised in a previous round of review and you feel that this manuscript is now acceptable for publication, you may indicate that here to bypass the “Comments to the Author” section, enter your conflict of interest statement in the “Confidential to Editor” section, and submit your "Accept" recommendation.

Reviewer #1: (No Response)

Reviewer #2: All comments have been addressed

2. Is the manuscript technically sound, and do the data support the conclusions?

Reviewer #1: Yes

Reviewer #2: Yes

3. Has the statistical analysis been performed appropriately and rigorously? 

Reviewer #1: I Don't Know

Reviewer #2: Yes

4. Have the authors made all data underlying the findings in their manuscript fully available?

Reviewer #1: Yes

Reviewer #2: Yes

5. Is the manuscript presented in an intelligible fashion and written in standard English?

Reviewer #1: Yes

Reviewer #2: Yes

6. Review Comments to the Author

Reviewer #1: Great job on the edits and additions, especially as a result of scrutinizing the youth weight data- you found some significant correlations that add valuable insight into the pathogenesis of hypoplasia.

Page 5 line 72-77: The anatomical examples you are citing have another obvious potential mechanism of interference- latch/milk removal, that is not acknowledged--while the less likely possibility that these anatomical variations reflect glandular tissue is the implication left, especially as the next sentence mentions physiological changes that may reflect internal functioning. Is this your intent?

Page 6 line 110: I believe you mean to say the proportion of women with low supply who also have hypoplasia markers, rather than the proportion of women in general... perhaps amend this to "the proportion of this population who also have one or more various anatomical..." or similar

Page 22- bottom-- Pre-pregnancy BMI section appears twice?

Page 31: ‘The high proportion of women self-reporting low supply in this sample with various proposed breast hypoplasia markers cannot be determinative until compared to a reference population of women with normal milk production/general population.’ I am not sure you need “general population” here, or perhaps this needs some tweaking.

Comment: P 23 version 1 discussion: for future research, might I suggest adding the variable of percentage of milk produced to relate to the anatomical variables? Kuznetsov used the following for degrees of hypogalactia: I - milk deficit is less than 25% of daily volume; II - milk deficit is 26-50%; III - milk deficit is 51-75%; IV - milk deficit is greater than 75% [3]. Kuznetsov, V. (2017). Clinical and pathogenetic aspects of hypogalactia in post parturient women. Актуальні проблеми сучасної медицини: Вісник української медичної стоматологічної академії, 17(1 (57)), 305-307. Response:

Thank you for this suggestion. We would like to add the following new sentence at the end of the Discussion “It would also be valuable to assess the relationship between deficit in maternal milk production (e.g. 75% deficit of daily volume [ref]) and breast hypoplasia markers.” However, we unfortunately have not been able to source the journal article and hence cannot cite it. We are therefore wondering if you could help us source it?

*Article and translation uploaded to you.

Reviewer #2: I am satisfied with attention to the comments of the reviewers, as responded to by changes or explanations..

7. PLOS authors have the option to publish the peer review history of their article (what does this mean?). If published, this will include your full peer review and any attached files.

Reviewer #1: No

Reviewer #2: **Yes: **Virginia Thorley

---

## [Author Response · Author response to Decision Letter 1]

1 Feb 2024

Editor comments:

Comment: 1. Please review your reference list to ensure that it is complete and correct. If you have cited papers that have been retracted, please include the rationale for doing so in the manuscript text, or remove these references and replace them with relevant current references. Any changes to the reference list should be mentioned in the rebuttal letter that accompanies your revised manuscript. If you need to cite a retracted article, indicate the article’s retracted status in the References list and also include a citation and full reference for the retraction notice. 

Response: No changes to reference list.

Reviewer 1 comments:

Comment: Great job on the edits and additions, especially as a result of scrutinizing the youth weight data- you found some significant correlations that add valuable insight into the pathogenesis of hypoplasia.

Response: Thank you!

Comment: Page 5 line 72-77: The anatomical examples you are citing have another obvious potential mechanism of interference- latch/milk removal, that is not acknowledged--while the less likely possibility that these anatomical variations reflect glandular tissue is the implication left, especially as the next sentence mentions physiological changes that may reflect internal functioning. Is this your intent?

Response: Thank you for raising this point. We have added an additional sentence as follows: “It is unclear if the poor breastfeeding outcomes were directly related to the anatomical issues or due to difficulties with infant latching to the breast and effectively removing milk.” (Page 5, lines 75-78)

Comment: Page 6 line 110: I believe you mean to say the proportion of women with low supply who also have hypoplasia markers, rather than the proportion of women in general... perhaps amend this to "the proportion of this population who also have one or more various anatomical..." or similar

Response: Thank you. We have edited this sentence to make it clearer as follows:

Previously submitted manuscript: “Therefore, using an online survey of women self-identifying as having low milk supply, we aimed to estimate the proportion of women with various anatomical breast characteristics related to breast hypoplasia, assess the feasibility of maternal self-report of these characteristics and to explore breast hypoplasia risk factors.”

Revised manuscript: ““Therefore, using an online survey of women self-identifying as having low milk supply, we aimed to estimate the proportion of this sample of women with various anatomical breast characteristics related to breast hypoplasia, assess the feasibility of maternal self-report of these characteristics and to explore breast hypoplasia risk factors.” (P 6, line 111)

Comment: Page 22- bottom-- Pre-pregnancy BMI section appears twice?

Response: Thank you, the duplicated pre-pregnancy BMI section in the previously submitted ‘Revised manuscript with Track Changes’ has been removed.

Comment: Page 31: ‘The high proportion of women self-reporting low supply in this sample with various proposed breast hypoplasia markers cannot be determinative until compared to a reference population of women with normal milk production/general population.’ I am not sure you need “general population” here, or perhaps this needs some tweaking.

Response: Thank you. We agree and have removed ‘/general population’ in the revised manuscript. (P 28, line 541)

Comment: P 23 version 1 discussion: for future research, might I suggest adding the variable of percentage of milk produced to relate to the anatomical variables? Kuznetsov used the following for degrees of hypogalactia: I - milk deficit is less than 25% of daily volume; II - milk deficit is 26-50%; III - milk deficit is 51-75%; IV - milk deficit is greater than 75% [3]. Kuznetsov, V. (2017). Clinical and pathogenetic aspects of hypogalactia in post parturient women. Актуальні проблеми сучасної медицини: Вісник української медичної стоматологічної академії, 17(1 (57)), 305-307. 

Thank you for this suggestion. We would like to add the following new sentence at the end of the Discussion “It would also be valuable to assess the relationship between deficit in maternal milk production (e.g. 75% deficit of daily volume [ref]) and breast hypoplasia markers.” However, we unfortunately have not been able to source the journal article and hence cannot cite it. We are therefore wondering if you could help us source it?

*Article and translation uploaded to you.

Response: Thank you. We have included the additional sentence and reference at the end of the Discussion section. (P 29, lines 556-558)

Reviewer 2 comments:

Comment: I am satisfied with attention to the comments of the reviewers, as responded to by changes or explanations.

Response: Thank you.

---

## [Decision Letter · Decision Letter 2]

13 Feb 2024

Breast hypoplasia markers among women who report insufficient milk production: A retrospective online survey

PONE-D-23-19510R2

Dear Dr. Renee,

We’re pleased to inform you that your manuscript has been judged scientifically suitable for publication and will be formally accepted for publication once it meets all outstanding technical requirements.

Kind regards,

Gilbert Sterling Octavius

Academic Editor

PLOS ONE

Additional Editor Comments (optional):

Reviewers' comments:

Reviewer's Responses to Questions

**Comments to the Author**

1. If the authors have adequately addressed your comments raised in a previous round of review and you feel that this manuscript is now acceptable for publication, you may indicate that here to bypass the “Comments to the Author” section, enter your conflict of interest statement in the “Confidential to Editor” section, and submit your "Accept" recommendation.

Reviewer #1: All comments have been addressed

Reviewer #2: All comments have been addressed

2. Is the manuscript technically sound, and do the data support the conclusions?

Reviewer #1: Yes

Reviewer #2: Yes

3. Has the statistical analysis been performed appropriately and rigorously? 

Reviewer #1: I Don't Know

Reviewer #2: Yes

4. Have the authors made all data underlying the findings in their manuscript fully available?

Reviewer #1: Yes

Reviewer #2: Yes

5. Is the manuscript presented in an intelligible fashion and written in standard English?

Reviewer #1: Yes

Reviewer #2: Yes

6. Review Comments to the Author

Reviewer #1: (No Response)

Reviewer #2: I believe that in this revision the authors have reflected upon and adequately addressed the issues raised by the reviewers. Their paper makes an important contribution to the discussion of breast hypoplasia in relation to human lactation, and their comments on body conformation in adolescence are important. They have addressed the limitations of their paper and, importantly, made pertinent suggestions for further studies.

7. PLOS authors have the option to publish the peer review history of their article (what does this mean?). If published, this will include your full peer review and any attached files.

Reviewer #1: No

Reviewer #2: **Yes: **Virginia Thorley

---

## [Editor Report · Acceptance letter]

20 Feb 2024

PONE-D-23-19510R2 

PLOS ONE

Dear Dr. Kam, 

I'm pleased to inform you that your manuscript has been deemed suitable for publication in PLOS ONE. Congratulations! Your manuscript is now being handed over to our production team.

Kind regards, 

on behalf of

Dr. Gilbert Sterling Octavius 

Academic Editor

PLOS ONE